# An Analysis of the Toxicity, Antioxidant, and Anti-Cancer Activity of Cinnamon Silver Nanoparticles in Comparison with Extracts and Fractions of Cinnamomum Cassia at Normal and Cancer Cell Levels

**DOI:** 10.3390/nano13050945

**Published:** 2023-03-05

**Authors:** Y. G. El-Baz, A. Moustafa, M. A. Ali, G. E. El-Desoky, S. M. Wabaidur, M. M. Faisal

**Affiliations:** 1Biochemistry Department, Faculty of Agriculture, Cairo University, Giza, Egypt; 2Chemistry Department, College of Science, King Saud University, Riyadh, Saudi Arabia; 3Centre of Materials Physics, Department of Physics, Durham University, Durham DH1 3LE, UK

**Keywords:** cinnamon bark, cinnamon-silver nanoparticle, polyphenols, anti-cancer activity

## Abstract

In this work, the extract of cinnamon bark was used for the green synthesis of cinnamon-Ag nanoparticles (CNPs) and other cinnamon samples, including ethanolic (EE) and aqueous (CE) extracts, chloroform (CF), ethyl acetate (EF), and methanol (MF) fractions. The polyphenol (PC) and flavonoid (FC) contents in all the cinnamon samples were determined. The synthesized CNPs were tested for the antioxidant activity (as DPPH radical scavenging percentage) in Bj-1 normal cells and HepG-2 cancer cells. Several antioxidant enzymes, including biomarkers, superoxide dismutase (SOD), catalase (CAT), glutathione peroxidase (GPx), glutathione-S-transferase (GST), and reduced glutathione (GSH), were verified for their effects on the viability and cytotoxicity of normal and cancer cells. The anti-cancer activity depended on apoptosis marker protein levels (Caspase3, P53, Bax, and Pcl2) in normal and cancerous cells. The obtained data showed higher PC and FC contents in CE samples, while CF showed the lowest levels. The IC_50_ values of all investigated samples were higher, while their antioxidant activities were lower than those of vitamin C (5.4 g/mL). The CNPs showed lower IC_50_ value (55.6 µg/mL), whereas the antioxidant activity inside or outside the Bj-1 or HepG-2 was found to be higher compared with other samples. All samples execrated a dose-dependent cytotoxicity by decreasing the cells’ viability percent of Bj-1 and HepG-2. Similarly, the anti-proliferative potency of CNPs on Bj-1 or HepG-2 at different concentrations was more effective than that of other samples. Higher concentrations of the CNPs (16 g/mL) showed greater cell death in Bj-1 (25.68%) and HepG-2 (29.49%), indicating powerful anti-cancer properties of the nanomaterials. After 48 h of CNPs treatment, both Bj-1 and HepG-2 showed significant increases in biomarker enzyme activities and reduced glutathione compared with other treated samples or untreated controls (*p* < 0.05). The anti-cancer biomarker activities of Caspas-3, P53, Bax, and Bcl-2 levels were significantly changed in Bj-1 or HepG-2 cells. The cinnamon samples were significantly increased in Caspase-3, Bax, and P53, while there were decreased Bcl-2 levels compared with control.

## 1. Introduction

Nanotechnology is the newest branch of modern science and the most promising area of research. It helps scientists to use applied knowledge of science and technology to govern matter on the atomic and molecular level. Biological sources have been used for progress of ecological and consistent methodology for the synthesis of various nanomaterials [1]. The interaction of inorganic nanoparticles with biological structures is one of the most exciting areas of research in the modern field of nanotechnology [2,3,4]. Nanoparticles are usually the size of 1 to 100 nm and exhibit different characteristics based on their smaller size, distribution, and morphology compared with bulk materials of the original sources [5,6]. A variety of plant extracts have been tested as potential reductants in Ag nanosynthesis instead of toxic chemicals [7,8], which are used in chemical reduction, photoreduction in reverse micelles, and radiation chemical reduction [9]. In addition to being expensive, these methods involve hazardous chemicals which may pose serious health and environmental risks to mankind. Spices have been the spice of life for human beings since time immemorial. There is something very appetizing about the scent and pungency of herbs, so they have become one of the most indispensable ingredients in the preparation of food that is palatable. In addition, spices possess antibacterial and medical/health benefits [10]. *Cinnamomum zeylanicum* is a small, evergreen tropical tree. *Cinnamon Ceylon* refers to Sri Lanka, the country that originally produced *Cinnamomum zeylanicum* [11]. Among the most popular herbs used to spice food is cinnamon bark. Cinnamon bark, branches, twigs, and leaves all contain useful phytochemicals; however, the bark is the most commercialized part among them. Additionally, its processed products include essential oils, oleoresins, and food additives. They are widely used in the cosmetic, beverage, pharmaceutical, and food industries as well.

In traditional medicine, it has been used as an anti-cold treatment, an anti-diarrhea treatment, and to treat other problems of the human digestive system. *C. zeylanicum* bark is rich in terpenoids, polyphenols, and flavonoids, which categorizes it as an antioxidant material [12]. The bark has also shown remarkable pharmacological effects to cure diabetes (type II) and insulin resistance [13]. Several antimicrobial properties of cinnamon essential oils make them suitable preservatives for foods [14]. Compared with commercially available products, zinc oxide nanoparticles reinforced with cinnamon extract have possessed anti-cancer, antioxidant, and anti-inflammatory properties [15]. The nano-cinnamon capsule has also shown remarkable pharmacological effects to cure diabetics (type II) and insulin resistance [16]. Aminzadeh et al. [17] studied the anti-tumor activities of aqueous cinnamon extract on cell line 5637, while Das et al. [18] mentioned the application of the extracts of *C. cassia* for improving blood circulation and inhibiting platelet coagulation. The extracts of *Ceylon cinnamon* are useful α-glucosidase and pancreatic α-amylase inhibitors and are involved in modifying glucose production in the liver [19]. Cinnamon oil-loaded chitosan nanoparticles had higher physical stability and were found to be also more effective against breast tumors, and the mechanisms associated with cinnamon effects on such diseases are connected to carbohydrate digestion as well as the release of fatty acids [20].

In this current work, we have used cinnamon bark extract for the green synthesis of Ag-nanoparticles and investigated its chemical characteristics. To the best of our knowledge, there are no reports in the literature on the possible cytotoxic, apoptotic, antioxidant, and carcinogenic effects of this plant or its fractions on HepG-2, human hepatoma cells. Therefore, considering the importance of cinnamon bark and its phytochemical constituents, the present work was designed to assess the possible cytotoxic, oxidative stress, and carcinogenic potential of cinnamon Ag-nanoparticles (CNPs), cinnamon extracts (CE), and their fractions (HF, CF, EF, and MF) on Bj-1 normal cells and HepG-2 cancerous cells. The findings also suggest the strong activity of cinnamon samples as a natural candidate for chemoprevention or therapeutic agents.

## 2. Materials and Methods

### 2.1. Ethical Approval

The current research study was ethically allowed by the Institutional Animal Caring and Use committee (CU-IACUC) reviewers. The first semester of Cairo University began in September 2022, and the last semester is in September 2024.

### 2.2. Chemicals and Supplies

All the chemicals used in this work were of analytical reagent grade and procured from Sigma-Aldrich (Burlington, MA, USA) and Fluka (Buchs, Switzerland) chemical companies. Folin–Ciocalteu reagent, quercetin, and gallic acid standards were provided by Sigma-Aldrich Co. (St. Louis, MO, USA). AlCl_3_ hexahydrate, sodium carbonate, and CH_3_OH were collected from Fisher Scientific (Fair Lawn, NJ, USA). A Milli-Q system (Millipore Corporation, Bedford, NH, USA) was used to produced purified water.

### 2.3. Preparation of Cinnamon Zeylanicum Bark Extracts and Fractions

Cinnamon barks (Figure 1) were dried up at room temperature and mechanically crushed into powder form using an electrical grinder. The aqueous infusion of the sample was prepared by taking cinnamon bark crumb (10 g) with 100 mL of distilled water in an infusion pan, followed by heating at 90 °C for a time span of 20 min. Then, the mixture was filtered to obtain the cinnamon extract (CE). The ethanolic extract (EE) of the cinnamon samples was prepared by taking the moistened cinnamon bark (100 g) with 96% ethanol into a percolator, and then soaked with 96% ethanol at the ratio of 1:8 for 24 h. It was continuously extracted by percolation until complete exhaustion had taken place. Water baths were used to evaporate the solvent until a semisolid extract was obtained, which was referred to as EE.

Preparation of crude fractions: Sixty grams of grounded cinnamon bark material was extracted successively into chloroform, hexane, methanol, and ethyl acetate at room temperature. After this, the extracts were filtered individually and concentrated using a rotary evaporator (Rota vapor R-/; BÜCHI Labortechnik AG, Flawil, Switzerland) to produce the fractions of hexane (HF), chloroform (CF), ethyl acetate (EF), and methanol (MF), respectively, and the samples were stored at 20 °C until their analysis. Six extracts and fractions were prepared (CE, EE, CF, HF, EF, and MF) and all of them were screened for various potential activities.

### 2.4. Biosynthesis of CNPs

One mL CE and 50 mL of 1 mM aqueous silver nitrate (AgNO_3_) solution were mixed and kept at room temperature for 8 h to produce Ag nanoparticles following Sathishkumar, et al.’s methods [21]. On reduction of silver (Ag^+^) to its reduced form (Ag°), the solution color changes from yellowish to dark. The preparation and stability of synthesized CNPs in sterile distilled water was established with zeta potential, UV-Vis analysis, and transmission electron microscopy (TEM). The CNPs were centrifuged at 10,000 rpm for 30 min. To remove free proteins/enzymes that were not capping the Ag nanoparticles, the pellets were washed three times with DI water, and then dried at 60 °C to remove the free proteins/enzymes [22].

### 2.5. Characterization of CNPs

#### 2.5.1. Measurement of Zeta Potential and Zeta Size

Zeta potential and zeta size were determined by using Malverns Zetasizer according to the method of Honary and Zahir [23].

#### 2.5.2. UV-Vis Spectral Analysis

The CNPs were characterized by widely used UV-Vis spectroscopy (Thermo Electron-Vision pro Software V2.03). The reduction of pure Ag^+^ to Ag° was scrutinized by optimizing the pH from 4 to 9 and the UV-Vis spectrum was prepared by the sampling of aliquots (0.3 mL) of CNPs solutions and further diluting them in DI water up to the 3.0 mL mark. In the range of wavelengths from 100 to 700 nm, UV-Vis spectra were analyzed.

#### 2.5.3. Morphological Characterization of CNPs

A drop of the prepared samples of the aqueous solution of CNPs was placed on carbon-coated copper grids, the films on the transmission electron microscopy (TEM) (model S-3400-N, Hitachi, Tokyo, Japan), the grids were allowed to stand for 2 min, and the extra solution was separated using a blotting paper for drying the grid. The size distribution of the CNPs were assessed based on TEM micrographs [24].

### 2.6. Phytochemical Investigations

#### 2.6.1. Quantitation of Total Phenolics (PC) and Flavonoids (FC)

##### Sample Preparation

About 10–50 mg of the CE sample was dissolved in 5 mL CH_3_OH and sonicated for 45 min while the temperature was kept at 40 °C. Then, the sample mixture was separated out by centrifuge for 10 min at 1000× *g*. In an amber bottle, the clear supernatant was collected and stored for analysis.

##### Total PC Analysis

As described earlier [25], Folin–Ciocalteu reagent was utilized to determine the total PCs of the extracts. A Cary 50 Bio UV-Vis spectrophotometer was used by Varian to measure the samples against a reagent blank at 765 nm. To 0.2 mL of the sample, phenolic Folin–Ciocalteu’s reagent was added (1:1) along with DI water (0.6 mL). After 5 min, saturated Na_2_CO_3_ solution (8%w/v in water) was further poured to the mixture, and made up the volume of 3 mL with DI water. The sample mixture was then kept in the dark for 30 min. After centrifuge, the absorbance of blue color samples were measured at 765 nm. The PC content was measured based on the gallic acid equivalents (GAE/g) of dry plant material using the standard curve (5–500 mg/L, *Y* = 0.0027*x* − 0.0055, *R*^2^ = 0.9999). Triplicate measurements were performed for all the analyses.

##### Total FC Analysis

The reported aluminum chloride colorimetry was adopted for the quantification of the total FC of the target sample [26]. For this, quercetin-based standard calibration curve was used. Stock quercetin solution was prepared by dissolving 5.0 mg quercetin in 1.0 mL methanol, then the standard solutions of quercetin were prepared by serial dilutions using methanol (5–200 μg/mL). Standard quercetin solutions (5–200 g/mL) were prepared by serial dilutions by dissolving 5.0 mg quercetin in 1.0 mL methanol. The resultant quercetin solutions were individually mixed with 0.6 mL of 2% AlCl_3_ and incubated for 60 min at room temperature. At 420 nm, we measured the absorbance of the reaction mixtures against a blank. The total FC in the test samples was quantitated from the standard quercetin calibration plot (*Y* = 0.0162*x* + 0.0044, *R*^2^ = 0.999) and stated as the quercetin equivalent (QE/g) of dried plant material. Triplicate quantifications were made for all measurements.

### 2.7. Determination of Antioxidant Capacity as Radical Scavenging Activity Percentage of DPPH and IC_50_ of Cinnamon Samples

A total antioxidant capacity test was carried out using 50 μg/mL diphenyl-2- picrylhidrazyl (DPPH), then the both maximum wavelength and absorbance were obtained and used as control absorbance. We tested cinnamon samples with concentrations of 20, 40, 60, 80, and 100 μg/mL. Furthermore, vitamin C of concentration 2, 4, 6, 8, and 10 μg/mL were used as the comparative standard. All these standards were (0.5 mL) individually reacted with 3.5 mL of DPPH and based on Spectrophotometry Genesys 30 Vis absorbance readings; this reaction result was noted. Using these absorbance and concentration data, the percentage inhibition (%inhibition) of cinnamon samples and vitamin C were quantitated using the equation,
DPPH ∗ scavenging activity (inhibition %) = [(Ac − As)/Ac] × 100
where the absorbance of the DPPH solution is Ac and the absorbance of the samples is As. A linear equation was created by using the %inhibition of cinnamon samples and vitamin C samples. The IC_50_ levels of cinnamon samples and vitamin C were calculated based on the above linear equation. We collected data based on the results of a DPPH test which were further used to determine total antioxidant capacity. In addition, GraphPad Prism V.0.9 was used in the experimental research data.

### 2.8. Cytotoxicity and Anticancer

American Type Culture Collection (ATCC, Manassas, VA, USA) provided all cell lines used in this study. The study used hepatocellular carcinoma (HepG-2, ATCC HB-8065) and skin fibroblast BJ-1 (ATCC CRL-2522) human cancer cell lines. Corning 75 cm^2^ u-shaped canted neck cell culture flasks with vent caps (Corning, New York, NY, USA) were utilized to culture cell lines in DMEM/high glucose augmented with 10% FBS, 2 mM L-glutamine, and 1% penicillin/streptomycin. In the following step, sub-on fluent cultures (70–80%) were trypsinized (trypsin 0.05%/0.53 mM EDTA) and split according to the seeding ratio [27].

#### 2.8.1. Cellular Antioxidant Activity

According to the manufacturer’s instructions, total antioxidant potency of the materials was calculated using a cellular antioxidant assay kit (Abcam ab242300). HepG-2 and Bj-1 cells were seeded in 96-well plates and treated with 25 g/mL for each treatment compared with the respective control. A cell-permeable DCFH-DA fluorescence probe dye was incubated after 24 h of treatment, and the bioflavonoid quercetin served as a control. After 30 min of incubation time, the cells were washed, and a free radical initiator was added to it to initiate the radical generation. The non-fluorescent DCFH-DA was converted to highly fluorescent DCF by the addition of a free radical initiator. The scavenging for free radicals increases with higher antioxidant potency, which inhibits the formation of DCF in a concentration-dependent manner. Thus, in a standard microplate fluorometer, fluorescence is measured over time, and antioxidant values are calculated as follows:The % in vivo antioxidant activity = [(Fc − Fs)/Fc] × 100
where Fc and Fs are the fluorescence of DCF and the sample, respectively. The antioxidant activity was determined by comparing antioxidant values to quercetin within the cell.

#### 2.8.2. 3-(4,5-Dimethyl-2-thiazolyl)-2,5-diphenyltetrazolium Bromide (MTT) Assay for Cell Viability, Proliferation, and Cytotoxicity

The 100 µL of medium/well in 96-well/plates (Hi media) were used for plating the cells (1 × 10^5^/well) into them. Confluence was reached by the cells after 48 h of incubation. Cinnamon samples were then added to RPMI-1640 media containing a variety of concentrations (0.5, 1.0, 2.0, 4.0, 8.0, and 16.0 g/mL). Following removal of the sample solution and washing with phosphate-buffered saline (pH 7.4), 20 L (5.0 mg/mL) of 0.5% MTT phosphate-buffer saline was added to each well. After 4 h of incubation, a mixture solution of 0.04 M HCl/isopropanol was added to the well. Viable cells’ absorbance was determined at 570 nm with reference to 655 nm using a microplate reader manufactured by Bio-Rad, Richmond, (CA, USA) using wells, while the cells containing no samples were considered as blanks. All experimental readings were collected in triplicates. Cinnamon samples were assessed for their effect on cancer cell proliferation using the following formula [28]:Cytotoxicity % = 100 − A570 of treated cells/A570 of control cells × 100%.

### 2.9. Determination of Cellular Oxidative Stress Enzymes

In order to evaluate the effect of cinnamon samples on the activity of HepG-2 and BJ-1, cells were seeded in RPMI-1640 medium containing 25 mg/mL of each cinnamon sample, with one 106 cell per flask, and incubated for 48 h at 37 °C under a humidified atmosphere of 5% CO_2_. After this, the cell medium was converted to serum-free medium (SFM) comprising 10 µL/mL of yogurt extract. Then, trypsin 0.05%/0.53 mM EDTA solutions were used to trypsinize the cell cultures after incubation. After washing in PBS, the cells were centrifuged at 2000 rpm for 5 min at 4 °C and resuspended in 1 mL PBS containing 0.1% Triton X-100. Using a 1.5 mL micro centrifuge tube, cells were sonicated twice for 10 s at 100 Hz (Vibra-cell, Sonics & Material) and centrifuged for 30 min at 4 °C at 14,000 rpm. The cells were sonicated for 2 min at 100 Hz in a 1.5 mL micro centrifuge tube placed on ice, then centrifuged at 14,000 rpm for 30 min at 4 °C after the sonication (Vibra-cell, Sonics & Material). Enzyme activity was tested in the supernatant of the tube. We measured the enzyme activity of SOD, CAT, GSH, and GPx in cell culture according to the manufacturer’s instructions for colorimetric kits ab65354, ab83464, ab142044, and ab102530 Abcam.

### 2.10. Determination of p53, Bax, Caspase-3, and Bcl-2 Protein Levels

The levels of the apoptosis markers p53, Bax, caspase-3, and Bcl-2 were determined 24 h after cinnamon doses of 25 g/mL were applied to cells. For 24 h, HepG-2 and Bj-1 cells were seeded at a concentration of 2 × 10^3^ cells/well in 6-well plates. Treatment media were replaced and the cells were incubated for an additional 24 h. After collecting the cells, they were lysed and centrifuged at 4 °C for 20 min at 10,000 rpm. A Bradford protein assay was used to determine the protein concentration in the supernatant [29]. For 1 h in the dark, 50 mg of total protein was incubated with 5 mL of caspase substrate in 100 mL of the reaction buffer. According to the manufacturer’s instructions, caspase-3 activity was assessed using a microplate reader at 405 nm using an Abcam colorimetric assay kit (AB39401) [11]. Following manufacturer instructions (ab207225, ab119506, and ab199080; Abcam), ELISA (enzyme-linked immunosorbent assay simple step) was used to measure the levels of apoptotic markers p53, Bcl-2-associated X (Bax), and B-cell lymphoma-2 (Bcl-2) in cell lysate [30].

### 2.11. Statistical Analysis

The Costal statistical package was used to analyze the data. Results are expressed as mean ± standard deviation (SD). ANOVA was used to verify both the significance of the difference parameters between mean values and the analysis of variance. Different letters within each column indicate significant differences at *p* ≤ 0.05 as detected by Duncan’s multiple range tests.

## 3. Results and Discussion

### 3.1. Zeta Potential and Zeta Size

Based on Figure 2 and Figure 3, their zeta potential is −12.3 (mV), indicating stability of CNPs with 201 (d.nm) size distributions. Zeta potential is a physicochemical parameter that influences the stability of nanoformulation. When zeta potentials are extremely positive or negative, they cause large repellent forces, whereas when their electric charges are similar, repulsion prevents particle aggregation, which in turn ensures easy redispersion. We performed the DLS in the water medium so it could become hydrated, and the increase the size was due to the hydrophilicity or agglomeration of nanoparticles, which can be seen in the TEM results (6–35 nm) [31]. Dynamic filtration analyzer (DFA) particle with a zeta potential greater than +30 mV or greater than −30 mV is also considered to be stable, as stated by Honary and Zahir [23].

### 3.2. UV-Visible Spectroscopy

Color change and UV-Vis spectroscopy were used to evaluate the synthesis of CNPs. CNPs absorb and scatter light very efficiently. Upon being excited by light at particular wavelengths, conduction electrons on metal surfaces undergo collective oscillations (surface plasmon resonance, SPR) that make them interact strongly with light. Synthesized CNPs exhibit higher absorption and scattering intensities compared with non-plasmonic nanoparticles of the same size due to SPR [32]. Within one hour, the reaction mixture turns yellowish-brown, and after eight hours, it becomes dark brown.

The color is ascribed to the reduction of Ag^+^ to Ag^0^ that activates SPR. CNPs solutions have a single sharp SPR band at 405 nm in their absorption spectra (Figure 4). In silver solution, a narrow plasmon absorption band is most prominent between 325 nm and 570 nm. At 405 nm, such a distinct peak could be seen, suggesting silver reduction as reported earlier [33]. In the absorption spectrum of CNPs solutions, there was a surface plasmon absorption band with a maximum of 446 nm, coinciding with the silver plasmon absorption band (325–525 nm).

### 3.3. Electron Microscopy

Figure 5 shows images of CNPs solutions obtained by electron microscopy. The results indicate that NPs adsorb and/or deposit on the surfaces of roughly sphere-shaped polydispersed particles. There are three different shapes of CNPs in the images: spheres, triangles, and irregularities. The spherical nanoparticles with a preferred growth direction along the Ag growth direction can be seen in Figure 5 as a typical example of ring patterns in the selected area electron diffraction. The average size of CNPs ranged from 6.0 to 37.0 nm for *C. zeylanicum* bark extract.

### 3.4. Polyphenol and Flavonoid Content in Cinnamon Extracts and Their Fractions

Hepatic damage can be effectively treated by applying medicinal plants high in antioxidant compounds [12]. There is a close correlation between the content of phenolic compounds in plant extracts and their antioxidant activity [31]. It has been suggested that flavonoids, which exist naturally in plants, can have positive effects on human health [14]. The anti-inflammatory, anti-cancer, anti-bacterial, and anti-allergic activities of flavonoid derivatives have been demonstrated in various research. In addition to this, flavonoids have also been shown to be highly efficient oxidant scavengers [13]. Table 1 lists the concentrations of polyphenols and flavonoids in CE, EE, and its fractions (HF, CF, EF, and MF). The total polyphenol content was highest in CE (79.43 mg GAE/g) and was followed by CE > EE > MF > EF > CF > HF. A higher concentration of polyphenols was found in both CE and EE, while lower concentrations of polyphenols (4.53 mg GAE/g) were found in HF and CF. There was a decreasing order of flavonoid contents: CE > EE > MF > EF > CF. The highest concentration of total FC (24.63 mg QE/g) was found in CE, while the lowest concentration (0.65 QE/g) was found in CF (Table 1, Figure 6 and Figure 7). The results are in good agreement with those reported by Ervina et al., who mentioned that infusion extracts and fractions of cinnamon can produce varying amounts of polyphenols and flavonoids [34]. Consequently, the yield of infusions and extracts is affected by different preparation methods.

### 3.5. Antioxidant Activity and IC_50_ of Cinnamon Samples

The absorbance at 517 nm was recorded at each concentration level of cinnamon samples. A spectrophotometer was used to measure the absorbance and calculate the percentage of inhibition (Table 2, Figure 8). The *X*-axis reflects the concentration of the cinnamon samples and the *Y*-axis represents the percentage of inhibition. From these data, a linear calibration curve was prepared (Table 3) and the IC_50_ values for CE, CNPs, EE, CF, EF, and MF samples were noted. Similarly, a standard linear regression curve for vitamin C was constructed, with the *X*-axis representing vitamin C concentration and the percent inhibition represented by the *Y*-axis; the linear equations are Y = 6.934X + 12.52 and R^2^ = 0.9988 (Table 3). The IC_50_ value for standard vitamin C was found to be 5.40 μg/mL. Based on standard curve data for vitamin C, the value was R^2^ = 0.9988, and R^2^ for CE, CNPs, EE, CF, EF, and MF were found to be 0.9973, 0.9993, 0.9983, 0.9991, 0.9984, and 0.9965, respectively. All equations showed good linear data, indicating reliable linearity. Cinnamon bark samples and vitamin C were evaluated by calculating the IC_50_, which suggests their ability to reduce radicals in DPPH by 50%. The IC_50_ value (Table 3 and Figure 8) for vitamin C was 5.4 μg/mL, while for CE, CNPs, EE, CF, EF, and MF samples the values were 64.3, 55.6, 65.68, 70.32, 65.102, and 66.9 μg/mL, respectively. In comparison with vitamin C, cinnamon bark samples exhibit a lower antioxidant capacity based on their IC_50_ value. CNPs show lower IC_50_ (49.51 g/mL) than other cinnamon samples, indicating higher antioxidant activity, while CF samples have the highest IC_50_ (70.32 g/mL) compared to all cinnamon samples, indicating lower antioxidant activity. According to Latief et al.’s findings, the antioxidant capacity of cinnamon bark extract was 49.0 μg/mL [35]. Additionally, Prahasti et al. reported that cinnamon bark extract had a 193.139 mg/mL antioxidant capacity [36]. Based on the results and discussion of the reported research, it can be decided that cinnamon bark extract has a strong total antioxidant capacity with IC_50_ value of 64.3 µg/mL. There are numerous constituents of cinnamon bark extract, including phenolics, alkaloids, tannins, saponins, flavonoids, glycosides, terpenoids, quinones, coumarins, cardiac glycosides, and betacyanides.

### 3.6. Cellular Antioxidant Activity

At 25µg/mL (below IC_50_), CE, CNPs, and its fractions (CF, EF, and MF) were assessed for their potential to scavenge DPPH radicals either within or outside cells (Bj-1 and HepG-2). The data were expressed as scavenging activity percentages. As shown in Table 4, CNPs outside the cells displayed higher scavenging activity against DPPH radicals compared with other investigated samples (CE, EE, EF, MF, CF) and followed the decreasing order (23.561 ± 2.11 > 22.64 ± 2.10 > 21.33 ± 2.65 > 20.67 ± 1.62 > 19.84 ± 1.33 > 19.029 ± 1.34), respectively. The lowest antioxidant activity was showed by the CF sample (19.029 ±1.34). All other samples including CE, EE, or CNPs outside the cells showed higher anti-oxidant activities compared with cinnamon fractions (CF, EF, MF), Table 4.

As compared with Bj-1 normal cells treated with CE, CNPs, or cinnamon fractions, cellular antioxidant activity increased by 12.46% and 7.0%, respectively, in cells treated with CE or CNPs compared with related control samples (untreated cells). In Bj-1 cells, after 24 h, the highest cellular antioxidant activity (35.43%) was recorded when treated with CNPs, followed by CE, which achieved an antioxidant activity of 29.54 percent, while CF treatment displayed the lowest antioxidant activity (24.93%) compared with the control and other samples Table 4.

According to previous outside antioxidant data, CNP treatment increased cellular antioxidant activity more than other cinnamon samples; this may be due to its higher solubility and bioavailability in aqueous cellular environments [16]. Additionally, the IC_50_ value of CNPs showed lower value (55.60) than that of other cinnamon samples, as shown in Table 3, which indicates that the antioxidant capacity is inversely proportional to the IC_50_ value. The higher contents of polyphenols and flavonoids in cinnamon extracts (Table 1 and Figure 6 and Figure 7) might increase the antioxidant activity compared with cinnamon fractions. We have found that CE and EE contain the highest concentrations of polyphenols and flavonoids, whereas CF contains the lowest amount of flavonoids. This indicates that polyphenol and flavonoid contents are responsible for increasing the antioxidant activity [37]. The results of this study suggest that the standard preparation method for obtaining infusions (i.e., CE), which is heating cinnamon bark at 90 oC for 20 min, is sufficient and results in significant antioxidant activity, and the conversion of CE to CNPs increases cinnamon’s antioxidant power owing to its greater solubility and bioavailability in living cells than that of other cinnamon samples [16].

These results also indicated that the CE, CNPs, EE, and cinnamon fractions are compounds with substantial antioxidant activity and they are from the class of polyphenols including phenolic volatile oil, flavonoids, and tannin. As Brewer [38] noted, phenolic compounds from plants have demonstrated antioxidant activity in general. These compounds are comprised of phenolic acids (gallic acid, caffeic acid, protocatechuic acid, and rosmarinic acid), phenolic diterpenes (carnosol, rosmanol carnosic acid, and rosmadial), and phenolic volatile oils (eugenol carvacrol, thymol, and menthol), as well as polyphenols such as flavones, flavonols, isoflavones, catechins, and tannins [39].

### 3.7. Cytotoxicity and Cell Viability Percent

There is a growing interest in establishing novel and effective treatment models for cancers such as hepatocellular carcinoma by exploiting the cytotoxic properties of natural compounds [40]. According to the results of the present study, CNPs, CE, and its fractions exhibit dose-dependent cytotoxicity on Bj-1 and HepG-2 cancerous cells (Table 5, Figure 9). These effects had higher potencies on HepG-2 cancerous cells than on Bj-1 normal cells. In the same respect, the anti-proliferative potency of CNPs and CE on Bj.1 or HepG-2 cells at different concentration levels was more effective than that of other cinnamon samples or fractions. With the higher concentration (16 g/mL), CNPs showed higher cell mortality (25.68%) of Bj-1 normal cells than that of HepG-2 cells (29.49%), while CF samples showed lower cell mortality of 15.1% and 16.2% for Bj-1 and HepG-2 cells, respectively, when compared with other cinnamon samples at this concentration. The CE sample showed higher cytotoxic effects on HepG-2 cancerous cells at different concentrations than EE or CF, EF, and, MF at the same concentration levels. These results indicate that CNPs and CE have powerful cytotoxicity against HepG-2 cancerous cells than against other cinnamon samples, and also increase the viability of Bj-1 normal cells at the lowest concentrations (0.5 and 1.0 µg/m) due to the higher content of polyphenols and flavonoids (Table 1, Figure 6 and Figure 7). Our results indicate that plants with high polyphenol and flavonoid contents have enhanced antioxidant activity, decreased IC_50_, and decreased cytotoxicity on cells.

### 3.8. Effects on Oxidative Stress Enzymes

CE, CNPs, EE, and cinnamon fractions (CF, EF, and MF) influence cellular antioxidant enzyme activity. In cell lysates of Bj-1 and HepG-2 after treatment with 25 µg/mL of cinnamon samples, the activity of SOD, reduced GSH, CAT, GPx, and glutathione-s-transferase (GST) was measured.

As compared with untreated cells, CE and CNPs significantly increased SOD, CAT, GSH, GPx, and GST levels in Bj-1 normal cells after 48 h (Table 6, Figure 10). A significant decrease in the enzymes’ activity was noticed with the CF compared with control, or other cinnamon samples treatment, while Bj-1 enzyme activities were not significantly altered by EE, EF, or MF samples.

The data in Table 6 show a significant increase (*p* > 0.05) in antioxidant enzyme biomarkers of Bj-1 cells under the influence of CNPs compared with other tested cinnamon samples. A significant increase in antioxidative enzymes was observed in HepG-2 cancerous cells treated with cinnamon extracts, CNPs, and cinnamon fractions, but CF treatment significantly decreased oxidative enzymes. CNPs showed significant increases in antioxidant enzymes in normal and cancer cells compared with control and other cinnamon samples. The findings suggest that CNPs have a powerful antioxidant effect and have the ability to reduce levels of oxidative stress in the cells, since they are more soluble and bioavailable [16].

On the other hand, several mechanisms are accountable for cinnamon’s effect on free radicals. By scavenging free radicals such as ROS and reactive nitrogen species (RON), it modulates the activity of catalase, GSH, and SOD that neutralize free radicals, and inhibits ROS-generating enzymes, including xanthine hydrogenase/oxidase and lipoxygenase/cyclooxygenase [41]. Cinnamon’s hydrophilic properties make it a good scavenger of peroxyl radicals, just like vitamin C [42].

### 3.9. Bj-1 and HepG-2 Treated Cell’s Poptotic Marker Protein Levels

In the present study, we have investigated the spectrum and modes-of-action of cinnamon samples against HepG-2 cancer cells, as well as compared their effects on Bj-1 normal cells. Among the six cinnamon samples, CNPs, CE, EE, and fractions (CF, EF, and MF) generated, only CNPs, CE, EE, and MF cinnamon bark fractions soluble in water exhibited potent anti-cancer activities as indicated by MTT cytotoxicity tests (Table 5). Apoptosis plays an important role in the development and health of a multicellular organism. In apoptosis, cells die in a controlled and regulated manner. Apoptosis has been shown to be a major pathway through which many medicinal and non-medicinal plants mediate anti-cancer effects. Apoptosis is regulated by two major pathways (intrinsic and extrinsic). As a result of both of these pathways, caspases enzymes act as death effector molecules in numerous types of cell death and converge to form a common pathway [27]. Generally, two types of caspases are involved in the regulation and execution of apoptosis: initiator caspases, which include caspases 2, 8, 9, and 10, and effector caspases, which include caspases 3, 6, and 7 [43].

Apoptosis is mediated by a number of genes besides caspases. The Bcl-2 family of proteins, including proapoptotic Bax and anti-apoptotic Bcl-2, regulate intrinsic pathways of apoptosis [40,41]. Apoptosis is regulated by proapoptotic Bax, which activates caspase initiators. Bax is one of the members of the Bcl-2 family that initiate apoptosis through the p53 gene, which functions as a tumor suppressor gene. The protein contents of p53, Bax, Bcl2, and Caspase-3 were calculated in Bj-1 and HepG-2 cells exposed to CE 25µg/mL, CNPs, EE, CF, EF, and ME (Table 7). No significant changes were observed on Caspas-3, P53, Bax, and Bcl-2 of Bj-1 normal cells due to CE or CNPs treatments, whereas EE, CF, EF, and MF treatments significantly decreased these apoptosis marker protein levels compared with control. On the other hand, HepG-2 cancerous cells treated with CE, CNPs, EE, or cinnamon fractions (CF, EF, MF) significantly increased Caspase-3, Bax, and P53 compared with control. However, all cinnamon samples significantly decreased Bcl-2 marker protein, except the CF sample, which showed no significant changes compared with control. Additionally, the data in the present study showed significant upregulation of pro-apoptotic caspase3, Bax, and p53 and down regulation of the anti-apoptotic BcL2 protein in HepG-2 cells treated with CNPs or CE after 24 h of incubation. CNPs treatment to HepG-2 exhibited highly significant changes in caspase-3, Bax, P53, and Bcl-2 levels compared with control or other treatments. According to the results obtained, CNPs and CE mediate their anti-cancer effects through apoptosis.

Several signal transduction pathways are reported to be involved in CNP’s potent anti-cancer activity, including pro-apoptotic (caspase3, P53, and Bax) and anti-apoptotic (Bcl-2). Based on Kerr et al.‘s findings, apoptosis is inversely associated with tumor progression, hyperplasia, and the formation of abnormal cells [44]. Many cancer cells are affected by cinnamon extract and its active compounds. In an in vivo melanoma model and in B16F10 cells, CE stimulated caspase-3 activity. However, the level of Bcl-2 was significantly decreased [45]. According to Sadeghi et al., cinnamon has an array of pharmacological properties, including antimicrobial, antioxidant, and anti-cancer properties [46]. Cancer is triggered and progressed by impaired apoptosis. There is increasing evidence that cinnamon, as a therapeutic agent, inhibits cancer cells by upregulating caspase3, P53, and Bax proteins and downregulating Bcl-2 proteins in apoptosis-related pathways.

### 3.10. Cellular Mechanism and Comparison of Works with Reported Literatures

In our earlier research study [43], the apoptosis related mechanism was discussed where the proteolytic activation of caspase-3 leads to DNA fragmentation and phosphatidylserine exposure causes degradation of nuclear protein and plasma membrane reversion [47]. The effects of plant-derived phytochemicals were suggested to be mediated by the induction of cell cycle arrest as well as apoptosis [28].

Ervina et al. [48] reported the antioxidant activity of Indonesian cinnamon bark for antioxidant activity. They reported that cinnamon bark infusion possesses the highest antioxidant activity, with IC_50_ value of 3.03. Additionally, the phytochemical analysis results indicated that polyphenol and phenolic volatile oil are the major antioxidant compounds. Ewyes at al. [49] reported in their research how fermentation effects the antioxidant and anti-cancer properties of Cinnamomum cassia. However, the authors do not report the preparation of nanoparticles and their application for various biological activities.

## 4. Conclusions

In the present study, CNPs, CE, and its fractions exhibit dose-dependent cytotoxicity on Bj-1 normal cells and HepG-2 cancer cells. Using CNPs significantly enhanced the anti-cancer and antioxidant activities of cinnamon samples on normal cells (Bj-1) or cancer cells (HepG-2), as they are more soluble and bioavailable. Additionally, as compared with control samples, all cinnamon samples increased the Caspase-3, Bax, and P53 levels, while decreasing Bcl-2 levels. Synthesized CNPs were found to be more effective as antioxidants and anti-cancer agents than other tested cinnamon samples. All these outcomes indicate that cinnamon has anti-cancer effects in cancer cells via apoptosis-related pathways by up- and down-regulating multiple proteins.

## Figures and Tables

**Figure 1 nanomaterials-13-00945-f001:**
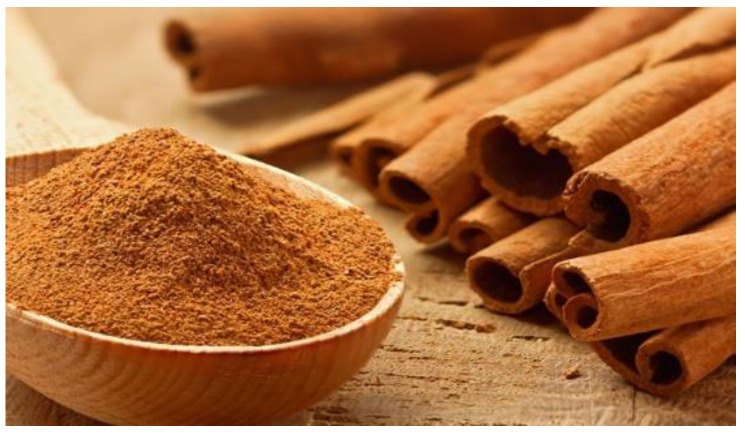
Cinnamon *Zeylanicum* barks and mechanically crushed cinnamon powder.

**Figure 2 nanomaterials-13-00945-f002:**
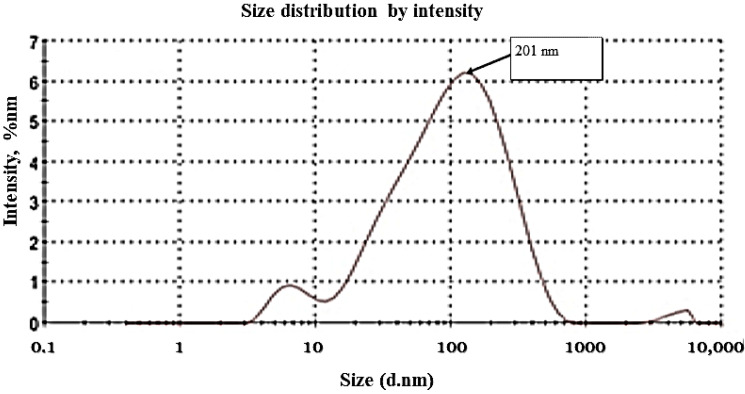
Zeta size of CNPs.

**Figure 3 nanomaterials-13-00945-f003:**
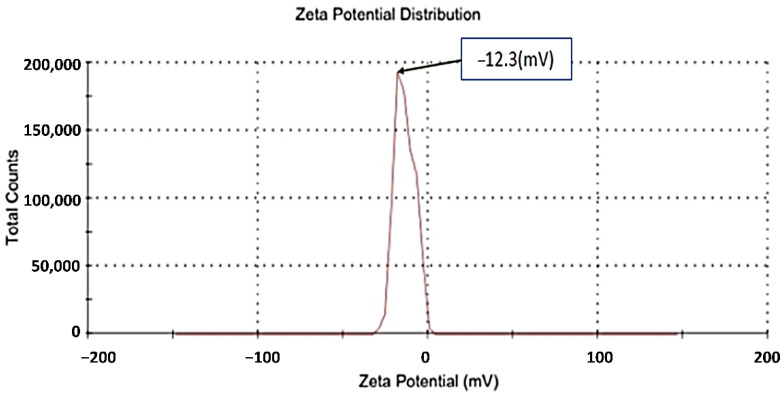
Zeta potential of CNPs.

**Figure 4 nanomaterials-13-00945-f004:**
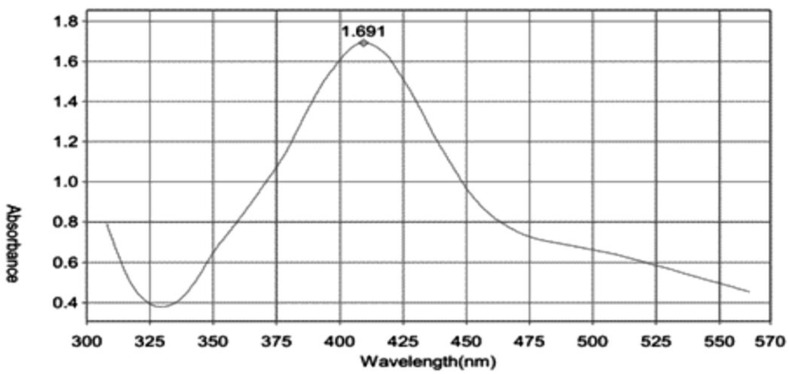
UV-Visible absorption spectra of synthesized CNPs, showing the surface plasmon resonance peak.

**Figure 5 nanomaterials-13-00945-f005:**
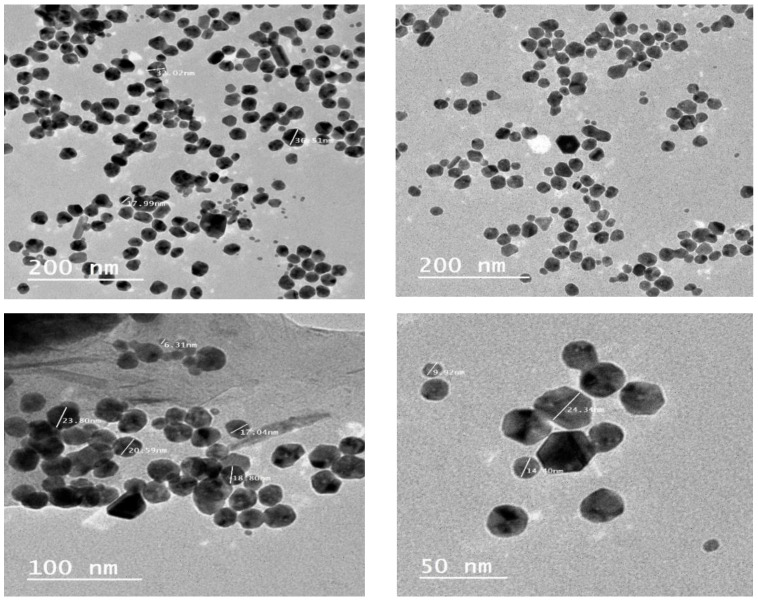
Crystalline clusters of CNPs.

**Figure 6 nanomaterials-13-00945-f006:**
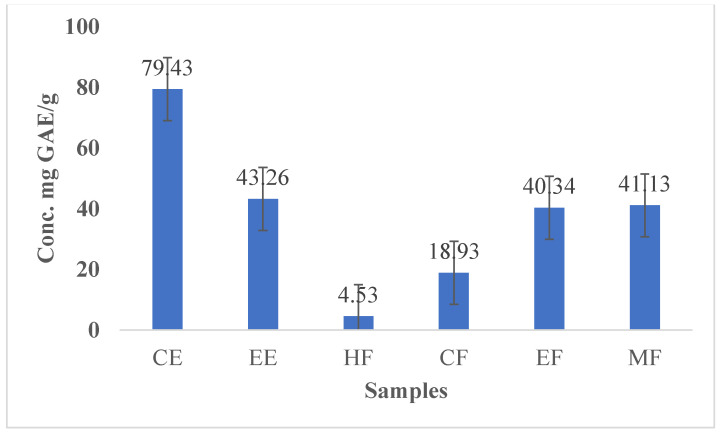
Total polyphenol content of CE, EE, and fractions (HF, CF, EF, and MF).

**Figure 7 nanomaterials-13-00945-f007:**
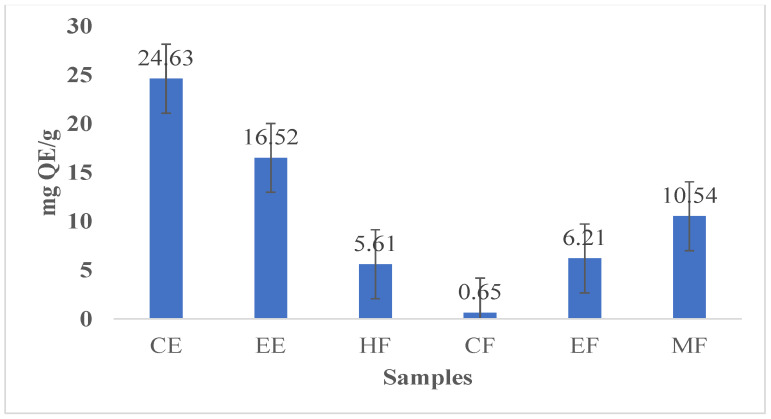
Total flavonoid content of CE, EE, and fractions (HF, CF, EF, and MF).

**Figure 8 nanomaterials-13-00945-f008:**
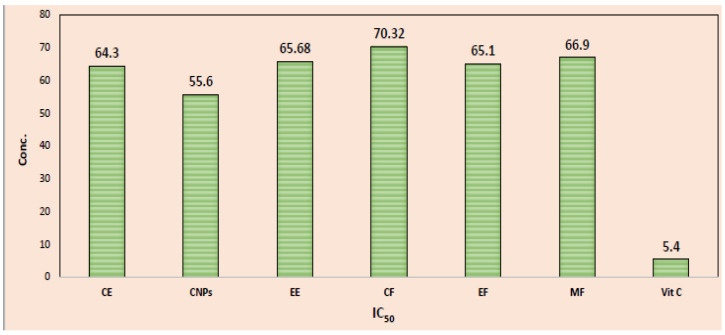
IC_50_ (The half maximal inhibitory concentration, which refers to the concentration that scavenges 50% of free radicals) of CE, CNPs, EE, CF, EF, MF, and vitamin C. RSA refers to radical scavenging activity. % DPPH radicals.

**Figure 9 nanomaterials-13-00945-f009:**
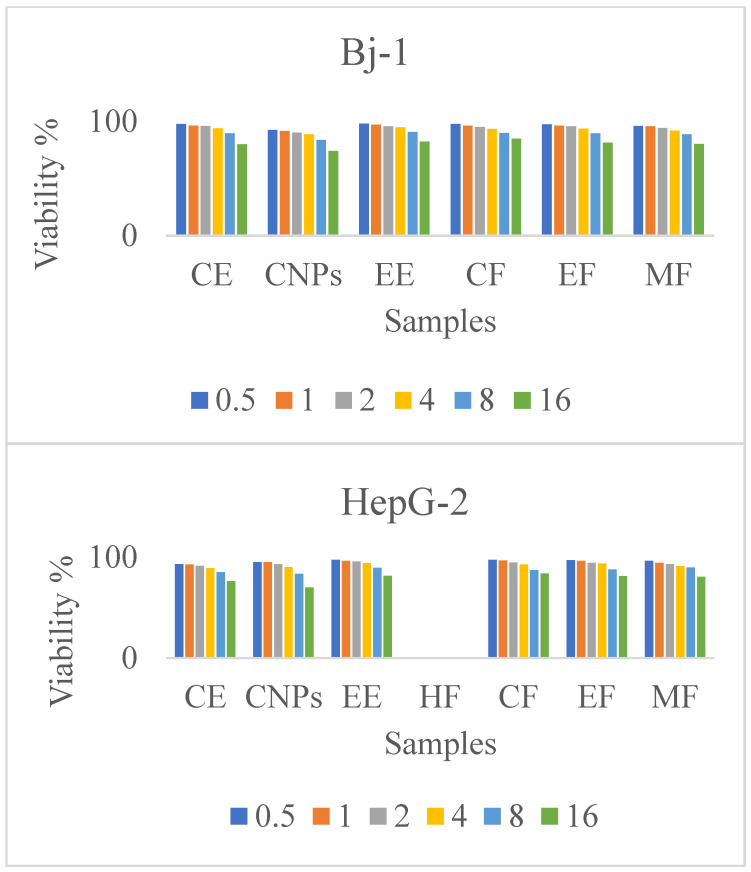
Viability % of Bj-1 HepG-2 cells.

**Figure 10 nanomaterials-13-00945-f010:**
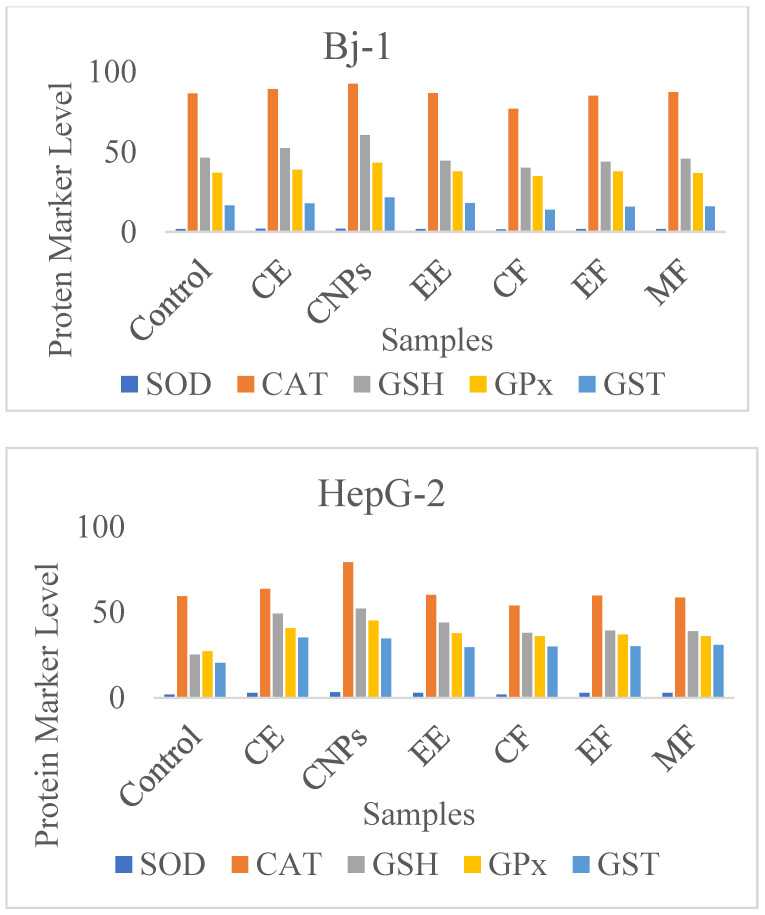
Oxidative stress enzyme activity in Bj-1 and HepG-2 cells.

**Table 1 nanomaterials-13-00945-t001:** Concentrations of PC and total FC in CE, EE, and its fractions (HF, CF, EF, and MF).

Cinnamon Samples	PC, mg GAE/g	FC, mg QE/g
CE	79.43 ± 18.87	24.63 ± 2.11
EE	43.26 ± 9.43	16.52 ± 1.21
HF	4.53 ± 2.32	5.61 ± 0.30
CF	18.93 ± 9.76	0.65 ± 0.01
EF	40.34 ± 11.9	6.21 ± 0.91
MF	41.13 ± 7.98	10.54 ± 1.01

Note: mean values ± standard error mg/g dry weight.

**Table 2 nanomaterials-13-00945-t002:** Concentrations, antioxidant capacity (% Inhibition) of cinnamon samples.

Conc. (µg/mL)	Antioxidant Capacity (% Inhibition)					
	CE	CNPs	EE	CF	EF	MF
20	18.532	19.971	17.511	15.605	16.861	15.902
40	32.732	34.331	31.731	29.285	31.561	30.442
60	46.932	48.691	45.591	42.965	46.261	44.982
80	61.152	63.051	60.171	56.672	60.961	59.522
100	75.332	77.44	74.391	70.325	75.661	74.062

**Table 3 nanomaterials-13-00945-t003:** Liner regression equations and IC_50_ of cinnamon samples.

Cinnamon Sample	Liner Regression Equations	r^2^	IC_50_ (µg/mL)
CE	Y = 0.710X + 4.332	0.9973	64.30
CNPs	Y = 0.718X + 5.611	0.9993	55.60
EE	Y = 0.711X + 3.291	0.9983	65.68
CF	Y = 0.684X +. 1.925	0.9991	70.32
EF	Y = 0.735X +. 2.161	0.9984	65.10
MF	Y = 0.727X + 1.362	0.9965	66.90
Vitamin C	Y = 6.934X. +. 12.25	0.9988	5.40

**Table 4 nanomaterials-13-00945-t004:** Inside and outside cellular antioxidant activity of CE, EE, CNPs, and cinnamon fractions (CF, EF, and MF) at concentrations of 25 µg/mL. Antioxidant activities were expressed as scavenging activity percentages of DPPH radicals.

Samples	%DPPH		
	Inside the Cell		Outside the Cell
	Bj-1	HepG-2	
Control	22.47 ± 0.54	21.52 ± 0.32	-------
CE	29.54 ± 0.65	33.52 ± 0.48	22.64 ± 0.76
CNPs	35.43 ± 0.59	44.87 ± 0.43	23.561 ± 0.72
EE	26.56 ± 0.55	27.63 ± 0.66	21.33 ± 0.56
CF	24.93 ± 0.65	26.54 ± 0.47	19.026 ± 0.59
EF	27.54 ± 0.68	28.59 ± 0.63	20.67 ± 0.77
MF	28.76 ± 0.62	28.6 ± 0.59	19.84 ± 0.71

Note: Bj-1 Cell = skin fibroblast normal cells, HepG-2 Cell = liver cancer cells. Different letters within each column designate significant differences at *p* ≤ 0.05.

**Table 5 nanomaterials-13-00945-t005:** Viability % of Bj-1 normal cells and HepG-2 cancerous cells under different concentrations of CE, EE, CNPs, and fractions (CF, EF, and MF).

Cells	Samples	Concentrations, µg/mL					
		0.5	1.0	2.0	4.0	8.0	16.0
Bj-1	CE	97.81 ± 2.95	96.32 ± 2.14	95.98 ± 2.76	93.94 ± 2.85	89.71 ± 3.73	80.14 ± 2.43
	CNPs	92.56 ± 3.32	91.71 ± 2.74	90.36 ± 3.11	88.65 ± 2.26	83.71 ± 2.51	74.32 ± 2.45
	EE	98.21 ± 2.87	97.21 ± 3.54	95.87 ± 2.54	94.98 ± 2.33	90.86 ± 2.32	82.32± 4.21
	CF	97.81 ± 3.87	96.22 ± 4.21	95.32 ± 2.43	93.45 ± 3.21	89.84 ± 3.78	84.87 ± 3.87
	EF	97.65 ± 3.12	96.32 ± 3.31	95.78 ± 3.41	93.58 ± 2.54	89.62 ± 2.32	81.58 ± 2.11
	MF	96.12 ± 2.65	95.65 ± 2.54	94..32 ± 4.11	92.09 ± 3.12	88.76 ± 4.21	80.44 ± 3.54
HepG-2	CE	93.1 ± 3.22	92.64 ± 2.38	91.39 ± 4.91	89.21 ± 1.91	85.1 ± 1.21	76.32 ± 1.12
	CNPs	95.11 ± 2.12	94.93 ± 0.12	93.14 ± 2.32	90.03 ± 1.41	83.32 ± 1.22	70.06 ± 1.21
	EE	97.43 ± 2.33	96.31 ± 3.21	95.76 ± 2.54	93.87 ± 3.65	89.46 ± 3.43	81.37 ± 4.65
	HF	NE	NE	NE	NE	NE	NE
	CF	97.34 ± 2.54	96.54 ± 3.44	94.54 ± 2.54	92.54 ± 4.32	87.23 ± 4.76	83.87 ± 3.66
	EF	97.12 ± 4.32	96.32 ± 2.43	94.21 ± 4.43	93.65 ± 2.76	87.76 ± 3.54	81.15 ± 3.54
	MF	96.23 ± 4.11	94.23 ± 3.54	93.11 ± 4.55	91.21 ± 3.43	89.87 ± 3.43	80.50 ± 3.45

Note: mean values ± standard error, Bj-1 Cell = skin fibroblast normal cells, HepG-2 Cell = liver cancer cells. Media containing different concentrations (0.5, 1.0, 2.0, 4.0, 8.0, and 16.0 µg/mL) of CE, CNPs, EE, HF, CF, EF, and MF.

**Table 6 nanomaterials-13-00945-t006:** Oxidative stress enzyme activity (SOD, CAT, GSH, GPx, and GST) in Bj-1 and HepG-2 cells treated with CE, EE, CNPs, and cinnamon fractions (CF, EF, and MF) compared with control.

Cells	Sample	Enzymes, U/10^6^ Cells				
		SOD	CAT	GSH	GPX	GST
Bj-1	Control	1.90 a ± 0.31	86.58 a ± 2.73	46.36 a ± 2.00	36.94 a ± 2.30	16.54 a ± 0.39
	CE	1.99 b± 0.16	89.24 b ± 2.51	52.42 b ±2.65	38.94 b ± 2.11	17.95 b ± 0.31
	CNPs	2.12 c ± 0.17	92.56 c ± 2.74	60.54 c ± 2.44	43.23 c ± 2.82	21.53 c ± 0.23
	EE	1.90 a ± 0.09	86.87 a ± 2.65	44.54 a ± 1.65	37.76 a ± 2.31	17.97 a ± 0.11
	CF	1.67 d ± 0.04	76.98 d ± 3.11	40.21 d ± 2.21	34.88 d ± 2.11	13.98 d ± 0.71
	EF	1.89 a ± 0.16	85.12 a ± 2.87	43.87 a ± 2.21	37.85 a ± 2.76	15.87 a ± 0.51
	MF	1.88 a ± 0.06	87.54 a ± 2.43	45.87 a ± 3.11	36.87 a ± 3.11	15.94 a ± 0.12
HepG-2	Control	1.81 a ±2.23	59.39 a ± 3.12	25.27 a ± 2.00	27.15 a ± 1.21	20.42 a ± 1.22
	CE	2.94 b ± 0.49	63.62 b ± 3.33	49.33 b ± 2.54	40.67 b ± 2.12	35.21 b ± 0.99
	CNPs	3.18 c ± 0.63	79.3 c ± 3.64	52.16 c ± 2.44	45.17 c ± 2.11	34.69 b ± 0.93
	EE	2.85 b ± 0.12	60.21 a ± 1.92	43.98 b ± 2.12	37.74 d ± 1.33	29.56 d ± 1.00
	CF	1.82 a ± 0.13	53.98 d ± 2.34	37.87 d ± 2.11	35.92 d ± 1.45	29.98 d ± 0.91
	EF	2.86 b ± 0.22	59.87 a ± 2.33	39.32 d ± 1.31	36.94 d ± 1.32	30.11 d ± 0.72
	MF	2.87 b ± 0.16	58.65 a ± 2.56	38.98 d ± 1.24	35.91 d ± 1.44	30.90 d ± 0.81

Note: mean values ± standard error; different letters within each column indicate significant differences at *p* ≤ 0.05 as determined by Duncan’s multiple range tests.

**Table 7 nanomaterials-13-00945-t007:** Apoptosis marker protein levels in HepG-2 and Bj-1 cells treated with CE, EE, CNPs, and fractions (CF, EF, and MF) at a concentration of 25µg/mL compared with control.

Cells	AMP	Caspas-3 pg/mL	P53 pg/mL	Bax pg/mL	Bcl-2 ng/mL
	Samples				
Bj-1	Control	60.53 a ± 1.65	80.13 a ± 2.07	23.98 a ± 1.65	2.76 a ± 0.57
	CE	59.41 a ± 1.33	81.12 a ± 1.31	23.41 a ± 1.87	2.80 a ± 0.23
	CNPs	60.03 a ± 2.16	80.34 a± 1.66	24.81 a ± 1.78	2.81 a ± 0.66
	EE	55.87 b ± 2.21	77.76 b ± 2.32	24.65 a ± 1.11	2.17 b ± 1.11
	CF	54.87 b ± 2.76	78.21 b ± 2.54	19.98 b ± 1.98	2.22 b ± 1.04
	EF	55.43 b ± 1.87	78.94 b ± 2.44	21.78 b ± 1.02	2.35 b ± 1.01
	MF	55.87 b ± 1.65	77.54 b ± 2.43	22.98 b ± 1.32	2.33 b ± 1.08
HepG-2	Control	73.76 a ± 3.41	92.29 a ± 2.76	40.76 a ± 3.54	2.88 a ± 0.98
	CE	119.33 b ± 3.23	140.19 b ± 3.32	67.77 b ± 3.32	1.45 b ± 0.19
	CNPs	147.75 c ± 2.67	169.97 c ± 3.34	87.55 c ± 2.31	1.10 c ± 0.08
	EE	105.23 d ± 2.65	100.32 d ± 3.49	60.98 d ± 1.32	1.46 b ± 0.07
	CF	89.65 e ± 2.43	92.87 e ± 3.12	52.76 e ± 2.32	2.32 a ± 0.09
	EF	91.98 e ± 3.47	95.43 e ± 3.11	55.87 e ± 2.54	1.29 b ± 0.07
	MF	95.54 e ± 2.56	97.87 e ± 2.87	55.98 e ± 3.12	1.41 b ± 0.08

Note: AMP = apoptosis marker protein. Different letters within each column indicate significant differences at *p* ≤ 0.05 as determined by Duncan’s multiple range tests.

## Data Availability

Not applicable.

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
