# Peer review of "An Analysis of the Toxicity, Antioxidant, and Anti-Cancer Activity of Cinnamon Silver Nanoparticles in Comparison with Extracts and Fractions of Cinnamomum Cassia at Normal and Cancer Cell Levels"

_nanomaterials, 2023, doi:10.3390/nano13050945_

Round 1

Reviewer 1 Report

The reviewed manuscript investigates the toxicity, antioxidant and anticancer activity of cinnamon silver nanoparticles in comparison to extracts and fractions of Cinnamomum cassia at normal and cancer cell levels. Authors demontrated that CNPs were found to be more effective as antioxidants and anticancer agents than other tested cinnamon samples.

1.    The description of the abstract is too wordy and needs to be concise.

2.    The introduction needs a lot of enhancement. I think it needs to be extended to illustrate the novelty and the importance of the work, as well as some previous related studies in published literatures.

3.    The mechanism of some results is insufficient. A discussion section is required to improve the readability and clarity of the manuscript.

4.    The format of the references is inconsistent. Please write in current format of the Journal.

Author Response

  1. The description of the abstract is too wordy and needs to be concise.

Answer: Corrected accordingly in the text.

  1. The introduction needs a lot of enhancement. I think it needs to be extended to illustrate the novelty and the importance of the work, as well as some previous related studies in published literatures.

Answer: Corrected accordingly in the text

  1. The mechanism of some results is insufficient. A discussion section is required to improve the readability and clarity of the manuscript.

Answer: Mechanism and comparison with other results are included.

Answer: Cellular mechanism

In our earlier research study [45] the apoptosis related mechanism was discussed where the proteolytic activation of caspase-3 leads to DNA fragmentation and phosphatidylserine exposure causes degradation of nuclear protein and plasma membrane reversion [46]. The effects of plant-derived phytochemicals was suggested to be mediated by the induction of cell cycle arrest as well as apoptosis [47]. Ervina et al., [48] reported the antioxidant activity of Indonesian cinnamon bark for antioxidant activity. They report that cinnamon bark infusion possesses highest antioxidant activity, with IC50 value of 3.03. Also, the phytochemical analysis results indicated that polyphenol and phenolic volatile oil are the major antioxidant compounds. Ewyes at al., [49] in their research reported how fermentation effect the antioxidant and anticancer properties of Cinnamomum cassia. However, the authors do not report the preparation of nanoparticles and their application for various biological activities.

  1. The format of the references is inconsistent. Please write in current format of the Journal.

Answer: All references are formatted according to journal guidelines.

Reviewer 2 Report

Thank you for allowing me to review this manuscript. Overall, this is a clear and well-written manuscript. This study has some clinical significance. Nonetheless, the manuscript can be further improved and the following concerns should be adequately addressed. My detailed comments are as follows: a. Please add related information on the cellular mechanism behind the results. b. Please compare with the previous study, where are the differences of this study?

Author Response

My detailed comments are as follows:

  1. Please add related information on the cellular mechanism behind the results.

Answer: Cellular mechanism

In our earlier research study [45] the apoptosis related mechanism was discussed where the proteolytic activation of caspase-3 leads to DNA fragmentation and phosphatidylserine exposure causes degradation of nuclear protein and plasma membrane reversion [46]. The effects of plant-derived phytochemicals was suggested to be mediated by the induction of cell cycle arrest as well as apoptosis [47].

  1. Please compare with the previous study, where are the differences of this study?

Ervina et al., [48] reported the antioxidant activity of Indonesian cinnamon bark for antioxidant activity. They report that cinnamon bark infusion possesses highest antioxidant activity, with IC50 value of 3.03. Also, the phytochemical analysis results indicated that polyphenol and phenolic volatile oil are the major antioxidant compounds. Ewyes at al., [49] in their research reported how fermentation effect the antioxidant and anticancer properties of Cinnamomum cassia. However, the authors do not report the preparation of nanoparticles and their application for various biological activities.

Reviewer 3 Report

The research article entitled “An analysis of the toxicity, antioxidant and anticancer activity 1 of cinnamon silver nanoparticles in comparison to extracts and fractions of Cinnamomum cassia at normal and cancer cell levels” focused on the preparations of silver nanoparticles using Cinnamomum cassia fractions. They evaluated the antioxidant and anticancer activity of cinnamon silver nanoparticles (CNPs) and compared them with Cinnamomum cassia extract and its fractions. There are many grammatical and sentence errors in the article, and the language organization needs to be improved. For these reasons, I conclude that the paper is not suitable for its current form and is recommended for minor publication.

1.       Authors have mentioned the DLS size of CNPs as 61 nm but figure 2 is indicating the size is more than 200 nm. Whereas the size of CNPs from TEM showed as 6-35 nm. The authors need to explain why the hydrodynamic size is more than the TEM size.

2.       Authors need to perform XRD to confirm the formation of silver nanoparticles. Also, mention the ICCD database number of AgNPs.

3.       Authors also need to provide HR-TEM (5 nm Scale) showing the fringe.

4.       Also, provide the EDX of CNPs to confirm the purity of the nanoparticle.

5.       Typographic errors need to be corrected. The language and grammar used throughout the manuscript need to be improved

6.       Table 5 of the Viability % of Bj-1 normal cells and HepG-2 cancerous cells may be represented as a line or Bar graph for a better understanding of readers. Also, Table 6.

7.       All the experiments need to be conducted in Triplicate. Total polyphenol content (Figure 6) and Flavonoid content (Figure 7) need to be conducted in triplicate and the SE bar to be added. 

Author Response

  1. Authors have mentioned the DLS size of CNPs as 61 nm but figure 2 is indicating the size is more than 200 nm. Whereas the size of CNPs from TEM showed as 6-35 nm. The authors need to explain why the hydrodynamic size is more than the TEM size.

The typos error has been corrected. The DLS we do in the water medium, so it might become hydrated and the increase the size was due to hydrophilicity or agglomeration of nanoparticles which can be seen in TEM results (6-35 nm).

  1. Authors need to perform XRD to confirm the formation of silver nanoparticles. Also, mention the ICCD database number of AgNPs.

Answer: Thank you so for the valuable comments of the honorable reviewer. Actually, We only indicated the formation and presence of the CNPs in the extracted materials using zeta potential, Uv – Vis spectroscopy and electron microscopy. Hence, the physical and chemical characteristics were not studied in detail using XRD analysis.

  1. Authors also need to provide HR-TEM (5 nm Scale) showing the fringe.

Answer: Thank you so for the valuable comments of the honorable reviewer. Actually, we only confirm the formation and presence of the CNPs in the extracted materials using zeta potential, Uv – Vis spectroscopy and transmission electron microscopy (TEM). Hence, we don’t consider the HR-TEM analysis here.

  1. Also, provide the EDX of CNPs to confirm the purity of the nanoparticle.

Answer: Thank you so for the valuable comments of the honorable reviewer. Actually, We only indicated the formation and presence the CNPs in the extracted materials using zeta potential, Uv – Vis spectroscopy and electron microscopy. Hence, the physical and chemical characteristics were not studied in detail using EDX analysis.

  1. Typographic errors need to be corrected. The language and grammar used throughout the manuscript need to be improved

Answer: The manuscript has been thoroughly checked and corrected.

  1. Table 5 of the Viability % of Bj-1 normal cells and HepG-2 cancerous cells may be represented as a line or Bar graph for a better understanding of readers. Also, Table 6.

Answer: The Bar graphs have been added accordingly.

Figure 9. Viability % of Bj-1 HepG-2 cells.

Figure 10. Oxidative stress enzymes activity in Bj-1 and HepG-2 cells.

  1. All the experiments need to be conducted in Triplicate. Total polyphenol content (Figure 6) and Flavonoid content (Figure 7) need to be conducted in triplicate and the SE bar to be added. 

Answer: Triplicate quantifications were made for all measurements, we have already mentioned in the text.

Reviewer 4 Report

The manuscript entitled " An analysis of the toxicity, antioxidant and anticancer activity of cinnamon silver nanoparticles in comparison to extracts and fractions of Cinnamomum cassia at normal and cancer cell levels" written by El-Baz and co-authors report on cinnamon silver nanoparticles antioxidant, anticancer and toxicity properties tested on normal and cancer cells comparing with extract and fractions of cinnamon. Although the results are interesting and could have potential developments for cancer treatments the way the author have presented their results is not enough critically analysed\discussed in light of current research in the field. The article has not yet reached the standard level for a broad audience like that one of Nanomaterials. I would suggest a deep revision of all part of manuscript before resubmitting it.

In particular several major points and issues, needing a profound re-writing\revision, are listed below:

1) A thorough critical updating of Literature references is strongly suggested (eg by adding recent review or research articles of competitive nanoparticles such as silica or gold NPs) 

2) The manuscript is written in a rather poor English which severely hamper the acceptance of the paper in this current form: therby a deep English revision by mother tongue revisor is strongly reccomended in order to polish -thoroughly- the text from several typos, still present  during the reading, needing to be removed\corrected (eg: page 2, line 40: "correct "scintests" in  "scientists"; page 6, line 111: "Measurment.." should be "Measurement.. " and so on)

3) Authors need also to strongly improve introduction (eg comparing with oither NPs?) and discussion (eg pitfalls and caveats of AgNPs?) , following text templating allover the manuscript according to Nanomaterials rules (eg follow rigorously Table templating)

4) In order to enhance the understanding of their outcomes, authors are also suggested to increase the low quality of data presentation (of all figures, in a more professional way: tables with data not having correct  statistical errors, missing controls ect ) which reflect insufficient relevance for an immediate publication.

6) Use of other type of cancer cells, by comparing a different type of tumor\neoplasy is also suggested. 

Alltogether, I would therefore suggest authors to perform a thorough major revision before re-submission.

Author Response

1) A thorough critical updating of Literature references is strongly suggested (eg by adding recent review or research articles of competitive nanoparticles such as silica or gold NPs) 

Answer: Few new works have been included.

2 Sivasankarapillai, Vishnu Sankar, Nishkala Krishnamoorthy, Gaber E. Eldesoky, Saikh Mohammad Wabaidur, Md Ataul Islam, Ragupathy Dhanusuraman, and Vinoth Kumar Ponnusamy. "One-pot green synthesis of ZnO nanoparticles using Scoparia Dulcis plant extract for antimicrobial and antioxidant activities." Applied Nanoscience (2022): 1-11.

3 Sharmila, Mohamed, Ramasamy Jothi Mani, Chelliah Parvathiraja, Sheik Mohammed Abdul Kader, Masoom Raza Siddiqui, Saikh Mohammad Wabaidur, Md Ataul Islam, and Wen-Cheng Lai. "Photocatalytic Dye Degradation and Bio-Insights of Honey-Produced α-Fe2O3 Nanoparticles." Water 14, no. 15 (2022): 2301.4 Herbin, H. Basalius, M. Aravind, M. Amalanathan, M. Sony Michael Mary, M. Maria Lenin, C. Parvathiraja, Masoom Raza Siddiqui, Saikh Mohammad Wabaidur, and Md Ataul Islam. "Synthesis of silver nanoparticles using syzygium malaccense fruit extract and evaluation of their catalytic activity and antibacterial properties." Journal of Inorganic and Organometallic Polymers and Materials (2022): 1-13.

2) The manuscript is written in a rather poor English which severely hamper the acceptance of the paper in this current form: therby a deep English revision by mother tongue revisor is strongly reccomended in order to polish -thoroughly- the text from several typos, still present  during the reading, needing to be removed\corrected (eg: page 2, line 40: "correct "scintests" in  "scientists"; page 6, line 111: "Measurment.." should be "Measurement.. " and so on)

Answer: Corrected throughout the manuscript.

3) Authors need also to strongly improve introduction (eg comparing with oither NPs?) and discussion (eg pitfalls and caveats of AgNPs?) , following text templating allover the manuscript according to Nanomaterials rules (eg follow rigorously Table templating)

Answer: Corrected accordingly in the manuscript

4) In order to enhance the understanding of their outcomes, authors are also suggested to increase the low quality of data presentation (of all figures, in a more professional way: tables with data not having correct statistical errors, missing controls ect) which reflect insufficient relevance for an immediate publication.

Answer: Few tables has been converted to figures as well with error bars.

6) Use of other type of cancer cells, by comparing a different type of tumor\neoplasy is also suggested. 

Alltogether, I would therefore suggest authors to perform a thorough major revision before re-submission.

Answer: Thank you so much for the comments raised by the honorable reviewer, we will do the comparison in our next work.

Round 2

Reviewer 1 Report

Accept

Author Response

Answer: More details have been added in the introduction and spell checked throughout the manuscript. Aminzadeh, et.al. [17] study the antitumor activities of Aqueous Cinnamon Extract on 5637 Cell Line, while Das, et. al. [18] mentioned the application of the extracts of C.cassia for improving blood circulation and inhibiting platelet coagulation. The extracts of Ceylon cinnamon are useful α-glucosidase and pancreatic α-amylase inhibitors and are involved in modifying glucose production in the liver [19]. Cinnamon oil-loaded chitosan nanoparticles had higher physical stability and were found to be also more effective against breast tumors and the mechanisms associated with cinnamon effects on such diseases are connected to carbohydrate digestion as well as the release of fatty acids [20].

Reviewer 4 Report

Manuscript acceptable after minor revision (few typos still to be corrected: eg in the introduction "scintists" to be corrected in "scientists")

Author Response

Answer: More details have been added in introduction and typos and spell checked throughout the manuscript.

 Aminzadeh, et.al. [17] study the antitumor activities of Aqueous Cinnamon Extract on 5637 Cell Line, while Das, et. al. [18] mentioned the application of the extracts of C.cassia for improving blood circulation and inhibiting platelet coagulation. The extracts of Ceylon cinnamon are useful α-glucosidase and pancreatic α-amylase inhibitors and are involved in modifying glucose production in the liver [19]. Cinnamon oil-loaded chitosan nanoparticles had higher physical stability and were found to be also more effective against breast tumors and the mechanisms associated with cinnamon effects on such diseases are connected to carbohydrate digestion as well as the release of fatty acids [20].